# A Novel Medical Image Enhancement Algorithm for Breast Cancer Detection on Mammography Images Using Machine Learning

**DOI:** 10.3390/diagnostics13030348

**Published:** 2023-01-18

**Authors:** Hanife Avcı, Jale Karakaya

**Affiliations:** Department of Biostatistics, Hacettepe University School of Medicine, Sihhiye, Ankara 06230, Turkey

**Keywords:** mammography images, classification performance, pre-processing methods, machine learning, GLCM, GLRLM

## Abstract

Mammography is the most preferred method for breast cancer screening. In this study, computer-aided diagnosis (CAD) systems were used to improve the image quality of mammography images and to detect suspicious areas. The main contribution of this study is to reveal the optimal combination of various pre-processing algorithms to enable better interpretation and classification of mammography images because pre-processing algorithms significantly affect the accuracy of segmentation and classification methods. In this study, the effect of combinations of different preprocessing methods in differentiating benign and malignant breast lesions was investigated. All image processing algorithms used for lesion detection were used in the mini-MIAS database. In the first step, label information and pectoral muscle resulting from the acquisition of mammography images were removed. In the second step, median filter (MF), contrast limited adaptive histogram equalization (CLAHE), and unsharp masking (USM) algorithms with different combinations of the resolution and visibility of images are increased. In the third step, suspicious regions are extracted from the mammograms using the k-means clustering technique. Then, features were extracted from the obtained ROIs. Finally, feature datasets were classified as normal/abnormal, and benign/malign (two class classification) using Machine Learning algorithms. Test performance measures of the classification methods were examined. In both classifications made in the study, lower classification performance values were obtained when the CLAHE algorithm was used alone as a pre-processing method compared to other pre-processing combinations. When the median filter and unsharp masking algorithms are added to the CLAHE algorithm, the performance of the classification methods has increased. In terms of classification success, Support Vector Machines, Random Forest, and Neural Networks showed the best performance. It was found by comparing the performances of the classification methods that different preprocessing algorithms were effective in detecting the presence of breast lesions and distinguishing benign and malignant.

## 1. Introduction

Cancer is one of the most common causes of death in the world. Breast cancer is the most common type of cancer among women worldwide [1]. Early diagnosis of cancer is very important in the success of treatment. Therefore, imaging techniques have been developed to increase the possibility of early diagnosis of breast cancer. Various imaging methods, including magnetic resonance imaging (MRI), ultrasonography (US) and mammography, are used to diagnose breast cancer [2]. Among these methods, mammography is relatively inexpensive, simple, fast and widely used as a screening test for the early detection of breast cancer because, with the help of images obtained by mammography, even small changes in the breast that cannot be detected by manual examination are detected [3].

Microcalcifications are the earliest signs of breast cancer that can be detected using screening methods [2]. These masses in breast tissues are difficult to diagnose when using mammography, as they often show low contrast. Images are distorted by random errors, called noise, caused by environmental factors or image capture devices. Various algorithms have been developed to remove these unwanted noises in the original image and improve the image. Images can be enhanced with computer-aided systems (CAD) using different techniques such as medical image processing [4]. Today, medical image processing is one of the fastest-growing areas in the healthcare industry.

The purpose of image processing is to make medical images used in diagnosis and treatment processes more reliable and comprehensible. In the diagnosis of breast cancer, CAD with various image processing algorithms and statistical methods can be useful in determining the presence of a mass or distinguishing benign/malign lesions [2].

Image processing can be defined as a method of transforming an image from one form to another. With image processing, new images are obtained by applying various operations to digital images. Mathematical algorithms are applied to these digital image data using computers and software suitable for the purpose. In summary, image processing is the process of obtaining numerical values suitable for the intended target after applying various computer algorithms to the obtained images. All operations applied to images are applied to increase the quality of the image. There are systematic or unsystematic errors and noises in the image structures due to the problems arising from the image source. All pixels in the image are affected by these errors and noises. For this reason, these errors should be eliminated with the help of pre-processing algorithms for the images to be used in practice and to be more understandable. Image processing consists of five stages: Image pre-processing, segmentation, feature extraction, feature selection, and classification are five stages of the image process, respectively. 

Before applying any image processing algorithms, pre-processing algorithms are of great importance to improve the results obtained [5]. In the literature, various pre-processing algorithms have been proposed to overcome the problem of misdiagnosis in mammography images [5,6,7,8,9,10]. These algorithms include several methods such as image resizing, mean filter, median filter, Gaussian filter, wiener filter, un-sharp masking, histogram equalization, anisotropic diffusion, and contrast-limited adaptive histogram equalization (CLAHE). In these studies, it was emphasized that different pre-processing algorithms affect classification performances [6,7,8].

Ganvir et al. [6] investigated the effects of median filter, wiener filter, anisotropic filter and wavelet filtering and anisotropic filtering one by one to overcome problems such as unwanted noise and low contrast in mammography images. These filtering methods were compared for accuracy with measurements such as standard deviation (SD), peak signal-to-noise ratio (PSNR), signal-to-noise ratio (SNR), self-similarity index measure (SSIM), and Root Mean Square Error (RMSE). It is stated that the anisotropic diffusion algorithm with wavelet-based filtering outperforms other pre-processing methods.

Ramani et al. [7] used mean filter, median filter, adaptive median filter and wiener filter algorithms as pre-processing methods in their studies. They compared these filtering methods with the values of mean square error, peak signal-to-noise ratio, average distance, maximum difference, etc., which are objective picture quality measurements. It has been concluded that the quality of the images with the adaptive median filter applied is better than the other filtering methods. 

Al-Najdawi et al. [8] used single and double combinations of CLAHE, median filter and Gaussian filter methods to improve images and facilitate segmentation. Six different pre-processing combinations were applied and presented to the radiologists in order to reach the best method that can best improve the visual details according to the radiologists. The results showed that the combination of CLAHE & Median filter algorithms used together gives better results.

Each of the double and triple combinations of these pre-processing methods has been studied separately in the literature [6,7,8]. These pre-processing methods have a direct effect on the segmentation step and feature matrix. However, there is no study in which different combinations of these step-by-step pre-processing methods are used and classification performances are evaluated on machine learning methods. In this study, un-sharp masking (USM) as an image sharpening filter, median filter (MF) as an image smoothing filter, and CLAHE algorithms to increase contrast are discussed.

The aim of this study is to investigate the contribution of the use of different combinations of various image enhancement algorithms applied to mammography images to the classification performance of machine learning methods in order to increase the contrast of the images and reduce the noise. Within the scope of the study, after different image pre-processing methods, segmentation, region of interest (ROI), feature extraction, feature selection, and classification steps were used. In the image pre-processing step, different algorithms were examined and their success in separating normal/abnormal tissues were compared. In the next step, the effect of pre-processing algorithms in separating benign/malign tissues was investigated.

## 2. Materials and Methods

We used an open-access database in the study. Figure 1 shows the flow diagram of the process of preparing the dataset used in the study. 

In this study, Fiji-ImageJ [11], MedPic [12], MATLAB version R2017b [13] were used for image processing; R Studio [14] software was used to examine the performance of classification methods. We draw the plots in Figures by using the “ggplot2” package [15] in RStudio.

### 2.1. Description of Dataset

The open-access mini-MIAS database was used in the study [16]. Although it is an old data set, it is still widely used in the literature [5,6,7,9,10,17,18]. This database is reachable at (http://peipa.essex.ac.uk/info/mias.html (accessed on 2 June 2022)). This dataset consists of 322 digitized mammography images, including the right and left breast images of 161 patients. Images are in portable gray map (PGM). In the database, all available mediolateral oblique (MLO) views of the left and right breast are included. It has a size of 1024 × 1024 pixels and a gray level range of 0–255.

### 2.2. Pre-Processing

Some mammography images in the mini-MIAS database contain label information. Since these labels originating from the mammography device have a high-density value, they may cause false results from the images. Therefore, labels need to be cleared from mammography images. For this, thresholding, morphological operations, and filtering methods are used in pre-processing [19,20,21].

First, the images were converted to binary (black and white) images by the thresholding method. A binary image is obtained by making the pixels above the threshold value white and the pixels below it black. Erosion and dilation morphological operations are applied to delete regions containing labels and numbers in black-and-white images. These black and white images obtained later were used as a mask on the original image. We aimed to eliminate the areas outside the breast area in the images obtained with these processes. Then, filtering methods, one of the pre-processing methods, were used in mammography images to increase the image quality and improve the segmentation results. In this study, different combinations of CLAHE, median filter and un-sharp masking algorithms were used as the pre-processing method. First, the CLAHE algorithm was tested individually. Then some possible combinations of filters such as Median Filter&CLAHE (MF&CLAHE), Median Filter&Un-sharp masking (MF&USM), CLAHE&Un-sharp masking (CLAHE&USM), and Median Filter&CLAHE&Un-sharp masking (MF&CLAHE&USM) were created. Each of these combinations was applied to each image. Figure 2 shows the results of applying the Median Filter&CLAHE&Un-sharp masking algorithms on the mdb171 mammography image containing a malign mass. 

### 2.3. Segmentation and Region of Interest (ROI)

The next important step for mammography images that are cleared of labels by pre-processing methods is to extract ROIs by clearing the pectoral muscle from the images with the appropriate segmentation method [21,22]. In image processing, it is important to emphasize the ROI because these regions are the part of an image that we want to filter or somehow manipulate. Color, shape, texture, contrast, etc., are obtained from these regions. Computer-aided systems can be used in diagnosis by classifying according to features. The details of the k-means clustering method, which is preferred as a segmentation method in the literature and which we used in this study, are explained below. This method has been preferred because it does not require any prior knowledge and is better than other region enlargement techniques. 

k-means clustering is one of the segmentation techniques commonly used in image processing applications [17,21,23]. This algorithm is a well-known unsupervised clustering method [21]. This segmentation method provides a simple and easy way to divide the image into different regions through a predetermined number of clusters. It is a segmentation algorithm that determines whether neighboring pixels belong to the same pixel or region, after selecting an initial pixel or region belonging to the image of interest and dividing the image into different regions after creating pixel clusters. Mammography images can be divided into three main clusters, including pectoral muscle, breast tissue, and background. First, centroids K, which are the starting point, are defined, *n* for each cluster. A feature region is then determined for each center that groups similar pixels. The principle of the k-means clustering segmentation method is given by Equation(1):(1)$$J=\overset{k}{\underset{j=1}{\sum }}\overset{n}{\underset{i=1}{\sum }}||x_{i}^{\left(j\right)}-c_{j}||{}^{2}$$

According to the number of clusters determined based on random methods, the centers of the clusters are determined. Here, $||x_{i}^{j}-c_{j}||{}^{2}$ is the distance from the point $x_{i}^{\left(j\right)}$ to the center of the $c_{j}$ group. The assignment of each pixel to the nearest cluster is based on the Euclidean distance between the point $x_{i}^{\left(j\right)}$ and the center of the $c_{j}$ group. Thus, function *J* represents the similarity measure of n pixels (objects) for each cluster.

After the images obtained after the segmentation process are applied as a mask on the original images, ground truth images are obtained. An example of the segmentation algorithm is shown in Figure 3.

### 2.4. Feature Extraction and Selection

In image processing, feature extraction plays a crucial role: it enables the extraction of numerical information (features) from medical images that cannot be detected by observation using appropriate statistical algorithms. For this purpose, different features such as statistical, texture, morphological, and shape features can be extracted from the images. In this study, Gray Level Co-occurrence Matrix (GLCM) and Gray Level Run Length Matrix (GLRLM) features, which are widely used in texture analysis, were extracted from each ROI sample. After the image pre-processing and segmentation processes were applied to the mammography images, feature extraction was performed. In this study, GLCM and GLRLM methods were used as feature extraction techniques. Twenty-two features were extracted with the GLCM method and 11 features were extracted with the GLRLM method. These features were extracted from the ROI samples, GLCM, and GLRLM matrices in four different angles directions: 0°, 45°, 90°, and 135°. The feature matrix was obtained by taking the average of these extracted features [9]. Thus, with the help of this feature matrix, mammography images were converted into numerical data. 

The correlation matrix was examined for feature selection. Among the 33 features obtained, the features with a correlation above 0.90 were eliminated and a selection was made. The classification performances of each selected feature alone were examined with the area under the curve (AUC) values obtained as a result of ROC Analysis. Therefore, a set of nine features are extracted for training the classifier methods. Figure 4 summarizes four different directions for GLCM and GLRLM matrices.

### 2.5. Classification

These numerical data obtained as a result of image processing were used as input variables in classification methods. In this study, lesion images were classified as normal/abnormal tissue in the first stage and benign/malign tissue in the second stage, using feature sets extracted with different techniques. Support vector machine (SVM), random forest (RF), artificial neural network (ANN), k-Nearest Neighbour (k-NN), naive Bayes (NB), and decision tree (DT) use the 322 images of the left and right breasts from a mini-MIAS database for the testing and training purposes. The data set is divided into 70–30% training and test sets. Models were obtained by using the leave-one-out cross-validation (LOOCV) procedure. The performances of these models established for classification methods were evaluated according to accuracy, sensitivity, specificity, the area under the ROC curve (AUC), and F1 evaluation criteria [24].

## 3. Results

In this study, a mini-MIAS dataset containing 322 mammography images of 161 patients was used. Various pre-processing methods were applied to remove unwanted information such as noise, tag information, and digitization noise (some straight lines) in images from the MIAS database. As a pre-processing step, different combinations of image smoothing algorithm median filter, image sharpening filter un-sharp masking, and CLAHE algorithms were used. In the next step, ROIs were obtained with the k-means clustering algorithm.

A total of 33 features were obtained from the ROI samples using GLCM, and GLRLM techniques. Then, the correlation matrix of these features was examined. Feature selection was made by eliminating one of the variables with a correlation of 0.90 and above. These selected features are summarized in Table 1. The average of the AUC values of all features in the CLAHE, MF&CLAHE, MF&USM, CLAHE&USM, and MF&CLAHE&USM algorithms alone ranged from 0.746 (0.520–1.00). Some features have been shown to perform very well on their own.

The results of the normal/abnormal classification of the characteristics of the analyzed data set are given in Table 2. When the CLAHE algorithm is used as a stand-alone pre-processing method, the effect on the performance of the classification methods was found to be lower. When the median filter is used together with the CLAHE algorithm, the classification performances increased. When the MF&CLAHE combination is used, AUC values are higher than the MF&USM combination, excluding the k-NN classification method. With the combination of CLAHE&USM, higher classification performances were obtained compared to single and other double combinations. In the combination of MF&CLAHE&USM, when all three different pre-processing methods are used together, the classification performances obtained are very close to the CLAHE&USM combination. According to the results in Table 2, when we look at measures such as overall accuracy, sensitivity, AUC, F-measure, the performances of SVM, RF, ANN, NB, and DT classification methods are more successful than the k-NN method. The graph showing the AUC values for five different pre-processing combinations and six different classification methods is presented in Figure 5.

Benign and malign classification results of the obtained features are given in Table 3. As with the normal and abnormal classification results, the use of the CLAHE algorithm alone as a pre-processing resulted in poor performance of the classification methods. However, when the MF&CLAHE combination was used as a pre-processing method in the classification of benign and malign, the performance of the classification methods increased compared to the CLAHE algorithm alone. When the MF&USM combination is used, higher classification performances are obtained than the CLAHE algorithm but lower than the MF&CLAHE combination. The CLAHE&USM combination gave the best classification results. When the MF&CLAHE&USM combination is applied, the classification results are slightly lower than CLAHE&USM. According to the results in Table 3, when we look at measures such as overall accuracy, sensitivity, AUC, and F-measure, the performances of SVM, RF, ANN, NB, and DT classification methods are more successful than the k-NN method. The graph showing the AUC values of five different pre-processing combinations and six different classification methods according to the benign/malign classification is presented in Figure 6.

## 4. Discussion

The determination of suspicious areas in mammography images by computer-aided systems has been investigated by many researchers in the last two decades. In the studies conducted in the literature, it is emphasized that the pre-processing step is very important in the segmentation and feature extraction stages in determining the suspicious regions. Among the various pre-processing methods, CLAHE, median filter, and un-sharp masking algorithms are commonly used [7,8,17]. Rarely, different methods such as the Gaussian filter, mean filter, and Sobel gradient have been used as pre-processing algorithms [22]. In most studies in the literature, pre-processing algorithms have been used in a single or double form. There is no similar study in the literature for the scenarios discussed in this study. We planned this study to see the effects of using different combinations of pre-processing methods on the performance of classification methods.

Al-Najdawi et al. [8] used CLAHE, median filter, and Gaussian filter algorithms as pre-processing methods in their study. These three filtering methods, single and double combinations, were applied to mammography images taken according to mediolateral oblique view (MLO) and craniocaudal view (CC) techniques. As a performance measure, sensitivity was 96.2% and specificity was 94.4%. In this study, in addition to the CLAHE and Median filter pre-processing algorithms, we created different pre-processing combinations by applying the unsharp masking pre-processing method to see the effect of the un-sharp masking pre-processing method, which is different from the literature.

Tiedeu et al. [25] increased the contrast of images with the CEI (i, j) algorithm they developed in their study. They used the Gaussian filtering method to soften the original image. After applying these two pre-processing methods, a set of moment-based geometric features were extracted from the ROI samples obtained by applying the adaptive thresholding segmentation method on 66 enhanced mammography images. With these properties, the specificity was 87.77% and the sensitivity was 100%. Mohanty et al. [26] extracted 19 features from ROI samples according to GLCM, and GLRLM techniques and classified them according to the C5.DT algorithm. Accuracy was 93.6% and the area under the curve (AUC) was 99.5%. Punitha et al. [10] classified 45 features obtained with the help of GLCM, and GLRLM techniques according to the feed-forward back propagation neural network algorithm. Sensitivity was 98.1%, specificity was 97.8%. In this study, when we used CLAHE&USM and MF&CLAHE&USM algorithms for normal/abnormal tissue classification and MF&CLAHE, CLAHE&USM, and MF&CLAHE&USM algorithms for benign/malign tissue classification, the highest AUC value was 1.00, sensitivity 1.00, and specificity for nine features obtained using GLCM, and GLRLM techniques in the feature extraction step. It was obtained as 1.00. In general, the performance measures were represented with sensitivity and specificity. However, we also preferred to interpret the AUC values that combine both performance measures.

It has been seen in studies in the literature that SVM generally gives high results [9,18]. In our study, SVM, RF, and ANN methods generally performed better than k-NN, NB, and DT especially in CLAHE&USM, and MF&CLAHE&USM pre-processing combinations.

In the normal/abnormal tissue classification of our study, CLAHE, MF&CLAHE, and MF&USM, in the benign/malign tissue classification, CLAHE, and MF&USM pre-processing combinations were low, and in the CLAHE&USM, and MF&CLAHE&USM algorithms, high classification performance measures were obtained. However, it has been observed that the performance measures in CLAHE&USM and MF&CLAHE&USM algorithms are very high. This situation is not considered to be an over-fitting problem of classification methods because the short-run emphasis, long-run emphasis, and long-run low gray level characteristics obtained according to this GLRLM technique were found to be quite high when the AUC values were examined alone. In general, the obtained features have high performance. According to the studies, it is expected that the result of image processing will be extremely high, although it is rare [27].

In the literature, it has been observed that the median filtering method has a higher noise reduction success for an image that is heavily distorted by salt and pepper noise [8]. Therefore, in future studies, after adding salt and pepper noise to normal images and applying MF&CLAHE pre-processing methods and classification performances the normal/abnormal texture can be examined because there are differences between the pre-processing techniques that should be used according to the noise types in the images [28].

In the study, image processing steps were performed on 322 mammography images from 161 people obtained from the open-access database. The algorithms used can be applied to larger data sets. The classification performances of various deep learning models (DenseNet, AlexNet, VGG 16, etc.) can also be compared when there are many mammography images [29].

The type of breast cancer may differ by the cells in the breast that turn into cancer [30]. The classification performances of the algorithms used in the study for different types of breast cancer can also be examined. There is no breast cancer type and phenotype information in the mini-MIAS dataset we used. However, in future studies, it can be planned to work with data in light of this information.

## 5. Conclusions

Depending on the shape of the cells on mammography images, a mass displayed can be normal, benign, or malign. Before applying image processing algorithms to identify suspicious areas in mammography images, the quality of the images needs to be improved. In this study, different pre-processing combinations were applied to remove noise and unwanted objects such as tag information on mammography images. At the same time, classification performances of data mining methods in different pre-processing combinations were compared. When the CLAHE algorithm is used alone as a pre-processing method, it has been observed that it has a lower classification performance than other pre-processing combinations. In normal/abnormal tissue classification, higher classification performances were obtained in CLAHE&USM and MF&CLAHE&USM algorithms compared to other pre-processing combinations. In benign/malign tissue classification, the classification performances of MF&CLAHE, CLAHE&USM, and MF&CLAHE&USM algorithms are higher than other pre-processing combinations. 

When we look at the best algorithm, in terms of classification performance, it can be said that SVM, RF, ANN, NB, and DT classification methods are more successful than the k-NN algorithm in both classifications.

At the end of this study, the appropriate combination with high classification performances in mammography images is proposed for the pre-processing algorithms, which has a significant impact on all steps in image processing.

## Figures and Tables

**Figure 1 diagnostics-13-00348-f001:**
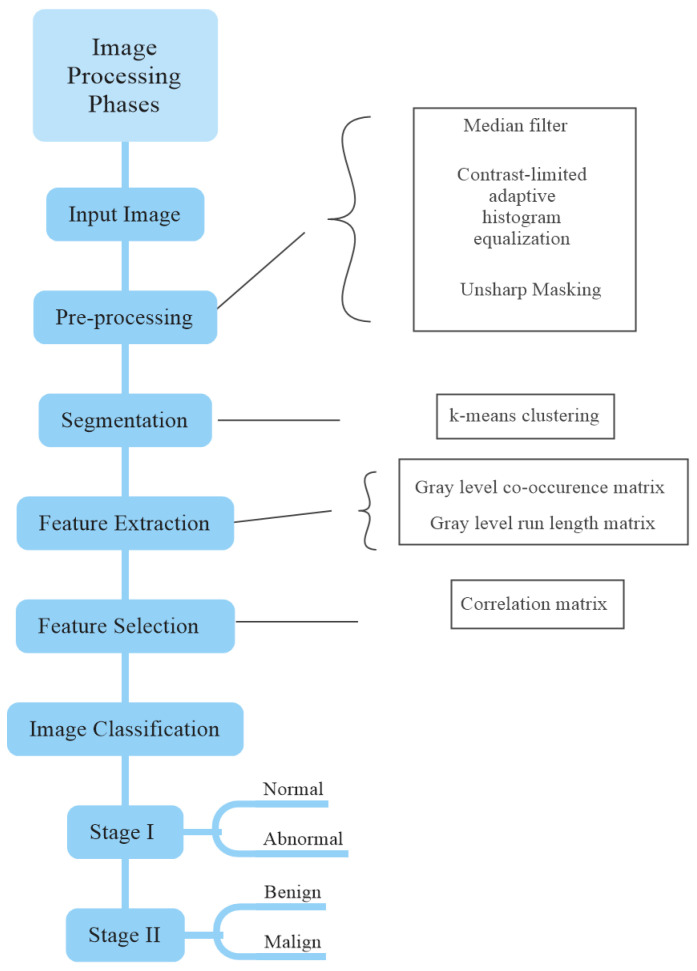
Flowchart of image processing.

**Figure 2 diagnostics-13-00348-f002:**
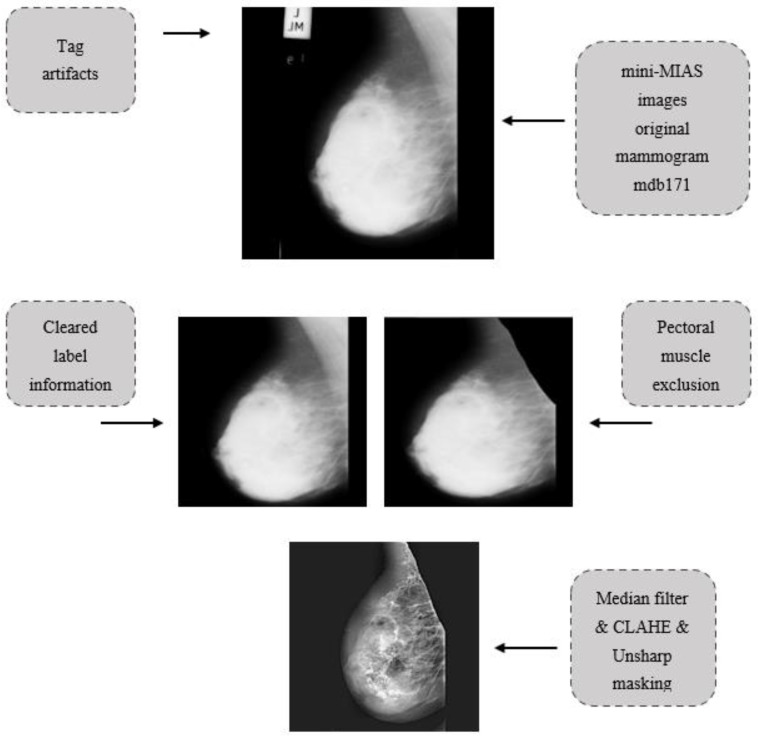
Schematic representation of the proposed triple pre-processing approach. While this schematic presentation started with a low-resolution image, the visibility and quality of the images increased with the proposed pre-processing methods.

**Figure 3 diagnostics-13-00348-f003:**
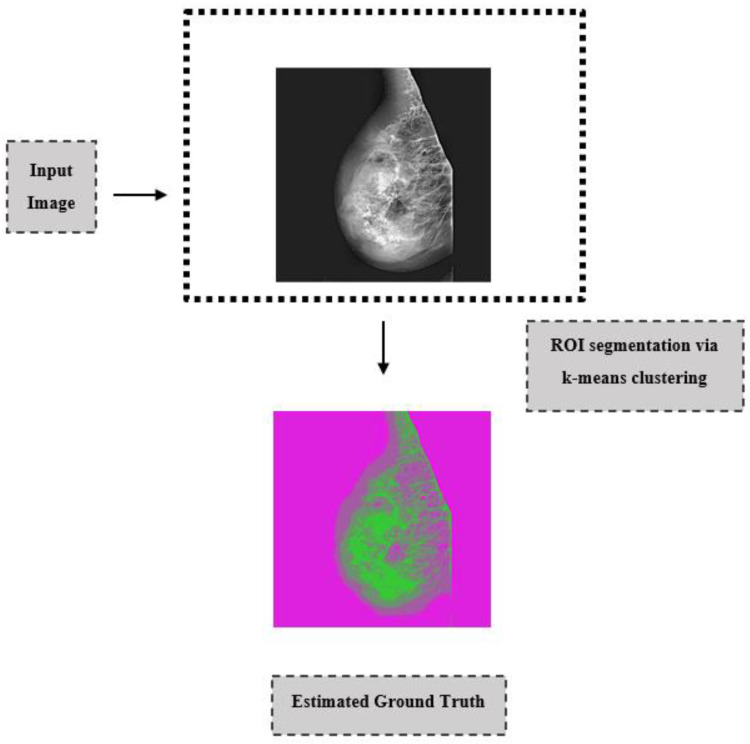
Segmentation examples for k-means clustering.

**Figure 4 diagnostics-13-00348-f004:**
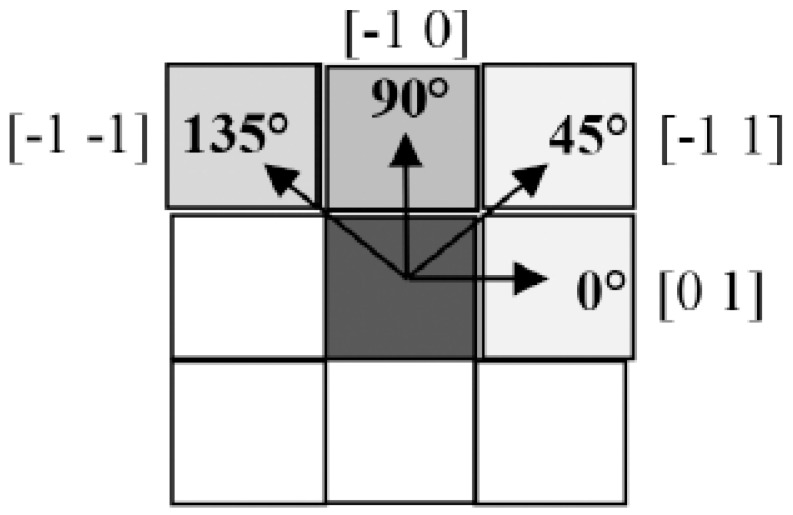
GLCM and GLRLM features for four angles from ROI.

**Figure 5 diagnostics-13-00348-f005:**
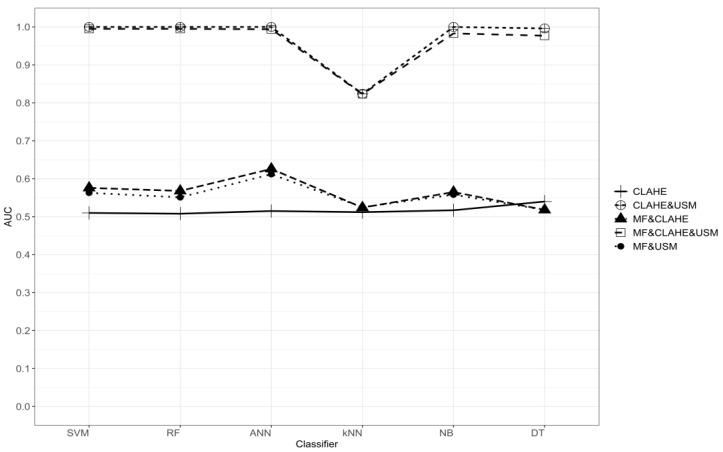
Comparison of the performances of classification methods with AUC values in different pre-processing combinations in normal and abnormal classification.

**Figure 6 diagnostics-13-00348-f006:**
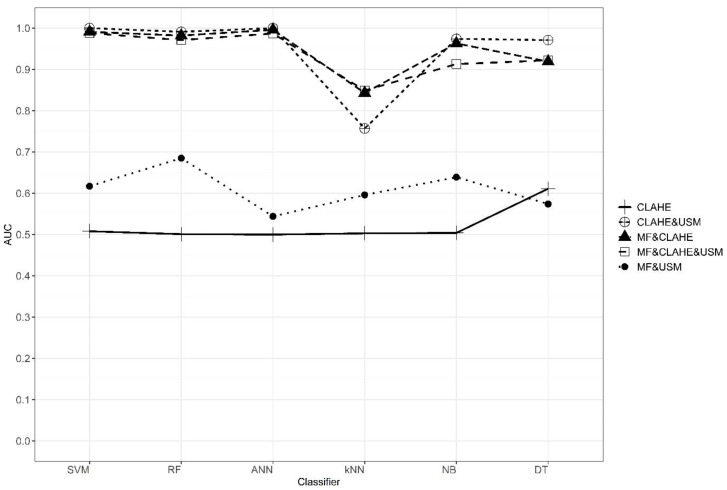
Comparison of the performances of classification methods with AUC values in different pre-processing combinations in benign and malign classification.

**Table 1 diagnostics-13-00348-t001:** List of nine features selected by using correlation coefficient.

Feature Matrix	Feature Name
GLCM	Autocorrelation
GLCM	Contrast
GLCM	Cluster prominence
GLCM	Entropy
GLRLM	Short run emphasis
GLRLM	Long run emphasis
GLRLM	Gray-level non-uniformity
GLRLM	Short run low gray-level emphasis
GLRLM	Long run low gray-level emphasis

**Table 2 diagnostics-13-00348-t002:** Performance of classification methods according to pre-processing algorithms for normal/abnormal classification.

		Performance Measures							
Classifier	Pre-Processing	Accuracy	Sensitivity	Specificity	PPV	NPV	AUC	BA	F1
**SVM**	CLAHE	0.649	0.035	0.912	0.143	0.689	0.510	0.473	0.055
	MF&CLAHE	0.639	0.281	0.815	0.429	0.697	0.576	0.548	0.339
	MF&USM	0.659	0.069	0.912	0.250	0.697	0.563	0.490	0.108
	CLAHE&USM	1.000	1.000	1.000	1.000	1.000	1.000	1.000	1.000
	MF&CLAHE&USM	1.000	1.000	1.000	1.000	1.000	0.995	1.000	1.000
**RF**	CLAHE	0.608	0.181	0.828	0.353	0.662	0.508	0.505	0.240
	MF&CLAHE	0.649	0.282	0.897	0.647	0.650	0.568	0.589	0.393
	MF&USM	0.659	0.355	0.803	0.458	0.726	0.551	0.579	0.400
	CLAHE&USM	1.000	1.000	1.000	1.000	1.000	1.000	1.000	1.000
	MF&CLAHE&USM	0.989	1.000	0.984	0.971	1.000	0.995	0.992	0.985
**ANN**	CLAHE	0.609	0.609	0.701	0.529	0.670	0.515	0.655	0.536
	MF&CLAHE	0.658	0.658	0.754	0.631	0.650	0.626	0.706	0.624
	MF&USM	0.665	0.665	0.760	0.640	0.730	0.612	0.713	0.633
	CLAHE&USM	1.000	1.000	1.000	1.000	1.000	1.000	1.000	1.000
	MF&CLAHE&USM	0.991	0.991	0.990	0.991	0.990	0.994	0.990	0.991
**k-NN**	CLAHE	0.618	0.304	0.716	0.250	0.768	0.512	0.510	0.274
	MF&CLAHE	0.598	0.210	0.847	0.470	0.625	0.524	0.529	0.291
	MF&USM	0.567	0.133	0.761	0.200	0.662	0.525	0.447	0.160
	CLAHE&USM	0.701	0.552	0.796	0.636	0.734	0.824	0.675	0.591
	MF&CLAHE&USM	0.742	0.697	0.766	0.605	0.830	0.823	0.731	0.648
**NB**	CLAHE	0.649	0.187	0.877	0.429	0.687	0.517	0.532	0.260
	MF&CLAHE	0.618	0.114	0.903	0.400	0.644	0.565	0.509	0.178
	MF&USM	0.732	0.400	0.881	0.600	0.766	0.558	0.640	0.480
	CLAHE&USM	1.000	1.000	1.000	1.000	1.000	1.000	1.000	1.000
	MF&CLAHE&USM	0.979	1.000	0.968	0.944	1.000	0.983	0.984	0.971
**DT**	CLAHE	0.526	0.054	0.817	0.154	0.583	0.540	0.435	0.080
	MF&CLAHE	0.557	0.289	0.729	0.407	0.614	0.518	0.509	0.338
	MF&USM	0.670	0.457	0.790	0.552	0.721	0.518	0.624	0.500
	CLAHE&USM	1.000	1.000	1.000	1.000	1.000	0.996	1.000	1.000
	MF&CLAHE&USM	0.969	1.000	0.957	0.900	1.000	0.977	0.978	0.947

**Table 3 diagnostics-13-00348-t003:** Performance of classification methods according to pre-processing algorithms for benign/malign classification.

		Performance Measures							
Classifier	Pre-Processing	Accuracy	Sensitivity	Specificity	PPV	NPV	AUC	BA	F1
**SVM**	CLAHE	0.470	0.167	0.812	0.500	0.464	0.508	0.489	0.250
	MF&CLAHE	0.970	1.000	0.944	0.941	1.000	0.991	0.972	0.969
	MF&USM	0.647	0.389	0.937	0.875	0.577	0.617	0.663	0.538
	CLAHE&USM	1.000	1.000	1.000	1.000	1.000	1.000	1.000	1.000
	MF&CLAHE&USM	0.941	0.937	0.937	0.937	0.944	0.988	0.941	0.937
**RF**	CLAHE	0.441	0.461	0.428	0.333	0.562	0.501	0.445	0.387
	MF&CLAHE	0.920	0.891	0.804	0.726	0.912	0.982	0.847	0.800
	MF&USM	0.647	0.565	0.722	0.643	0.650	0.685	0.642	0.600
	CLAHE&USM	0.970	0.941	1.000	1.000	0.944	0.991	0.970	0.969
	MF&CLAHE&USM	0.970	1.000	0.947	0.937	1.000	0.971	0.973	0.967
**ANN**	CLAHE	0.531	0.531	0.430	0.530	0.520	0.500	0.480	0.531
	MF&CLAHE	0.991	0.991	0.980	0.991	0.990	0.996	0.985	0.991
	MF&USM	0.602	0.602	0.725	0.600	0.605	0.544	0.663	0.590
	CLAHE&USM	1.000	1.000	1.000	1.000	1.000	1.000	1.000	1.000
	MF&CLAHE&USM	0.935	0.929	0.920	0.929	0.930	0.987	0.925	0.929
**k-NN**	CLAHE	0.471	0.333	0.545	0.286	0.600	0.503	0.439	0.307
	MF&CLAHE	0.794	0.583	0.909	0.778	0.800	0.843	0.746	0.667
	MF&USM	0.676	0.538	0.762	0.583	0.727	0.596	0.650	0.560
	CLAHE&USM	0.617	0.526	0.733	0.714	0.550	0.757	0.630	0.605
	MF&CLAHE&USM	0.823	1.000	0.667	0.727	1.000	0.849	0.833	0.842
**NB**	CLAHE	0.441	0.400	0.458	0.235	0.647	0.504	0.430	0.296
	MF&CLAHE	0.912	1.000	0.842	0.833	1.000	0.963	0.921	0.910
	MF&USM	0.647	0.500	0.778	0.667	0.636	0.639	0.639	0.571
	CLAHE&USM	0.971	1.000	0.941	0.944	1.000	0.974	0.971	0.971
	MF&CLAHE&USM	0.706	1.000	0.545	0.545	1.000	0.913	0.773	0.706
**DT**	CLAHE	0.471	0.636	0.391	0.333	0.692	0.611	0.514	0.437
	MF&CLAHE	0.882	0.923	0.857	0.800	0.947	0.919	0.890	0.857
	MF&USM	0.588	0.600	0.579	0.529	0.647	0.574	0.589	0.562
	CLAHE&USM	0.970	1.000	0.933	0.950	1.000	0.971	0.967	0.974
	MF&CLAHE&USM	0.941	1.000	0.888	0.888	1.000	0.922	0.944	0.941

## Data Availability

Images used to compare combinations of three different pre-processing methods as described in Section 2.1 are available at: http://peipa.essex.ac.uk/info/mias.html (accessed on 2 June 2022).

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
