# Peer review of "A Novel Medical Image Enhancement Algorithm for Breast Cancer Detection on Mammography Images Using Machine Learning"

_diagnostics, 2023, doi:10.3390/diagnostics13030348_

Round 1

Reviewer 1 Report

This study is fascinating and well-structured.

However, there are a few amendments I would recommend.

- There is a lack of citations in the introduction chapter. 

- Conclusions usually do not include discussion points and citations. I want to see the conclusions more to the point and answer the aim specifically.

Author Response

We are grateful and would like to thank for their valuable comments that will increase the quality of our manuscript. We carefully considered the issues and corrected them. Below, you may find the detailed response to previously addressed comments.

We hope that our corrections will fulfill the request given.

Sincerely yours,

Reviewer 2 Report

Differentiating different types of breast cancer and a different phenotype on them would be more helpful.  

This would also increase interest of the reader.

Overall a good approach.

Author Response

(The authors gave the same response as above.)
